# Pretrain then Adapt: Uncertainty-Aware Test-Time Adaptation for Text-based Person Search

## Abstract

Text-based person search faces inherent limitations due to data scarcity, driven by stringent privacy constraints and the high cost of manual annotation. To mitigate this, existing methods usually rely on a **Pretrain-then-Finetune** paradigm, where models are first pretrained on synthetic person-caption data to establish cross-modal alignment, followed by fine-tuning on labeled real-world datasets. However, this paradigm lacks practicality in real-world deployment scenarios, where large-scale annotated target-domain data is typically inaccessible. In this work, we propose a new **Pretrain-then-Adapt** paradigm that eliminates reliance on extensive target-domain supervision. The key underpinning our approach is Uncertainty-Aware Test-Time Adaptation (UATTA), a framework enabling dynamic model adaptation using only unlabeled test data, with minimal computational overhead. UATTA introduces a bidirectional retrieval disagreement mechanism to estimate uncertainty, *i.e.*, low uncertainty is assigned when an image-text pair ranks highly in both image-to-text and text-to-image retrieval, indicating high alignment; otherwise, high uncertainty is detected. This indicator drives test-time model recalibration without labels, effectively mitigating domain shift. We validate UATTA on four benchmarks, *i.e.*, CUHK-PEDES, ICFG-PEDES, RSTPReid, and PAB, showing consistent improvements across both CLIP-based (one-stage) and XVLM-based (two-stage) frameworks. Ablation studies confirm that UATTA outperforms existing test-time adaptation strategies, establishing a new benchmark for label-efficient, deployable person search systems.

## 1 Introduction

Text-based person search (Li et al., 2017; Zhu et al., 2021; Ding et al., 2021), which involves matching natural language descriptions to specific individuals within large-scale image galleries, is a critical task with applications ranging from locating missing persons (Bukhari et al., 2023) to enhancing smart city management (Khan et al., 2021; Zheng & Zheng, 2024). Unlike conventional image-based person re-identification (Zheng et al., 2015; 2017), this modality offers a more intuitive and accessible query interface for system operators (Zheng et al., 2020).

Despite its practical advantages, the efficacy of current methods is severely hampered by the domain shift problem, where models trained in controlled settings exhibit significant performance degradation when deployed in unseen, real-world environments. State-of-the-art approaches typically attempt to mitigate this challenge through a **Pretrain-then-Finetune** paradigm (Shao et al., 2023; Jiang & Ye, 2023; Nguyen et al., 2024; Tan et al., 2024; Jiang et al., 2025). This involves first pretraining on large-scale, often synthetic, person-caption datasets to establish robust cross-modal alignments, followed by fine-tuning on smaller, domain-specific annotated datasets such as CUHK-PEDES (Li et al., 2017). However, the reliance on labeled data for fine-tuning renders this paradigm impractical for many real-world deployments, owing to the scarcity of such data due to stringent privacy regulations (Gaikwad & Karmakar, 2023) and prohibitive annotation costs (Shao et al., 2023).

To this end, we propose an alternative **Pretrain-then-Adapt** paradigm, leveraging source-free test-time adaptation (TTA) (Wang et al., 2021; Dong et al., 2025) to adapt a pretrained model to a new target domain using only unlabeled test samples. A common TTA strategy is to update model

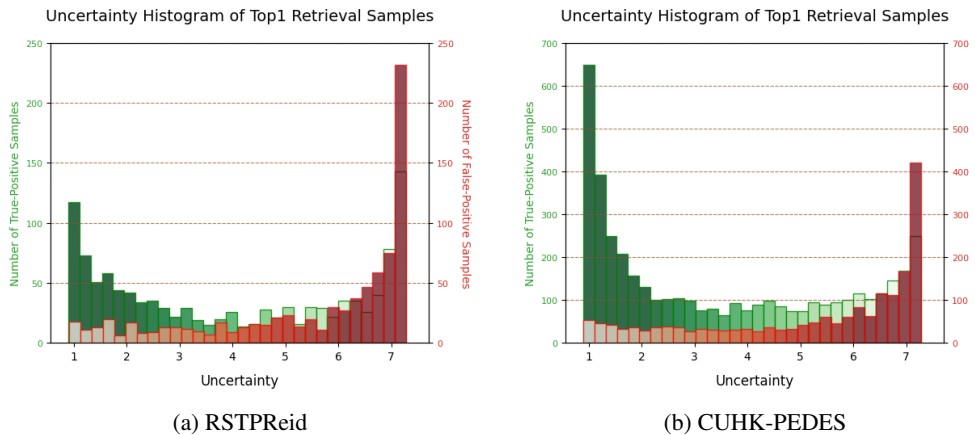

(a) RSTPReid                                    (b) CUHK-PEDES

Figure 1: **Statistical Overview of our Uncertainty Indicator**. We count the sample number according to the proposed uncertainty score for True Positives (TP) and False Positives (FP) on (a) RSTPReid (Zhu et al., 2021) and (b) CUHK-PEDES (Li et al., 2017). TP samples consistently cluster in the low-uncertainty region, while FP samples concentrate in the high-uncertainty region across both benchmarks. Therefore, it could serve as the indicator for the test-time adaptation.

parameters by minimizing an objective function, such as prediction entropy (Wang et al., 2021; Yang et al., 2022), to increase the model's confidence. This approach, however, presents a significant risk of error accumulation (Zhao et al., 2024); the model can become overconfident in its own erroneous predictions, reinforcing them during adaptation and converging to a suboptimal state. In the context of retrieval, this means the model indiscriminately amplifies the influence of false positives alongside true positives. This raises a crucial research question: *How can we mitigate the risk of overconfident, erroneous adaptation in cross-modal person retrieval, a task that demands fine-grained matching?*

In this work, we contend that a principled approach to this challenge lies in uncertainty-aware adaptation. We introduce the **Uncertainty-Aware Test-Time Adaptation (UATTA)** framework, which explicitly estimates and leverages prediction uncertainty to guide the adaptation process. Our core hypothesis is that retrieval uncertainty can be effectively gauged through *bidirectional retrieval disagreement*. A high-uncertainty match will exhibit incongruity between the text-to-image and the corresponding image-to-text retrieval directions, whereas a confident, low-uncertainty match will show symmetric alignment. This principle, illustrated in Fig. 1, allows us to identify and downweight potential false positives during adaptation. Specifically, for a one-stage retrieval model based on CLIP (Radford et al., 2021), we quantify uncertainty using the relative disparity between mutual recall probabilities derived from the Image-Text Contrastive (ITC) loss. This uncertainty measure is then used to rectify the entropy minimization objective by re-weighting it with the reciprocal of the uncertainty. For two-stage retrieval architectures like XVLM (Zeng et al., 2021), we apply the same bidirectional uncertainty principle to modulate the entropy of the fine-grained predictions from the Image-Text Matching (ITM) head. Our primary contributions are as follows:

- **A Practical Paradigm for Test-Time Adaptation.** We explore a **Pretrain-then-Adapt** paradigm for text-based person search that alleviates the need for labeled target-domain data. This framework offers a practical alternative to the standard Pretrain-then-Finetune pipeline, enhancing deployability in real-world scenarios where data annotation is infeasible.

- **An Uncertainty-Guided Adaptation Method.** We propose an Uncertainty-Aware Test-Time Adaptation (UATTA) framework designed to address domain shift under unsupervised conditions. The method introduces a **bidirectional retrieval disagreement** mechanism to estimate prediction uncertainty. This signal is used to guide the adaptation, aiming to curb error accumulation from overconfident predictions during the test-time optimization process.

- **Comprehensive Empirical Evaluation.** We conduct extensive experiments on four challenging benchmarks (CUHK-PEDES (Li et al., 2017), ICFG-PEDES (Ding et al., 2021), RSTPReid (Zhu et al., 2021), and PAB (Yang et al., 2024b)). Our results show that UATTA achieves consistent performance improvements over baseline methods across different model architectures.

The findings validate the efficacy of our uncertainty-guided approach and suggest it is a promising direction for label-free adaptation in this domain.

## 2 RELATED WORK

**Text-based Person Search.** Text-based person search aims to find the target person of interest via a text query. Different from image-based search, the text query is more intuitive for users. A typical dataset is CUHK-PEDES (Li et al., 2017). To align person images and text, recent works usually adopt a pretrain-then-finetune paradigm, in which models first establish cross-modal alignment on synthetic person-caption data and then fine-tune on limited real-world annotations. (Shao et al., 2023) apply CLIP (Radford et al., 2021) with a novel divide-conquer-combine strategy to automatically annotate pseudo-text descriptions for a large-scale person re-identification image dataset (Fu et al., 2021), which reduces human labor and cost. With the help of image generative models, (Yang et al., 2023) collect a new large-scale cross-modal dataset MALS (Yang et al., 2023), containing real-world text descriptions and corresponding generated person images with multiple attributes, providing an alternative for real-world person privacy via automatic image generation and attribute extraction. Following this synthetic-pretrain and real-world-finetune approach, (Tan et al., 2024; Jiang et al., 2025) boost text-based person search performance by exploiting Multi-modal Large Language Models to obtain text descriptions with various language structures and styles. Existing test-time inference pipelines of this paradigm can be divided into one-stage CLIP-based (Radford et al., 2021) and XVLM-based (Zeng et al., 2021) frameworks. The former (Jiang & Ye, 2023; Shao et al., 2023; Tan et al., 2024; Jiang et al., 2025) extracts vision and language features independently via separate single-modal models and predicts alignment based on Image-Text Contrastive (ITC) similarity (Radford et al., 2021). The latter (Li et al., 2021; 2022; 2023; Zeng et al., 2021; Yang et al., 2023; 2024b) employs an additional fine-grained cross-modal interaction module to exploit Image-Text Matching (ITM) learning and predict binary matching results to rectify top-$K$ results from the first stage. In this paper, we propose a universal Pretrain-then-Adapt paradigm that is not constrained by the scarcity of annotated labels for both one-stage and two-stage frameworks.

**Test-Time Adaptation.** Test-time adaptation (TTA) has emerged as a promising paradigm for domain shift mitigation without source data access. Parameter-metric approaches (Wang et al., 2021; Yang et al., 2022) minimize prediction entropy through lightweight parameter updates, e.g., Batch-Norm (Ioffe & Szegedy, 2015) statistics. However, these approaches suffer from confirmation bias as domain shift induces high-confidence errors, a phenomenon exacerbated in cross-modal retrieval where false positives deteriorate performance (Zhao et al., 2024). Memory-based approaches (Iwasawa & Matsuo, 2021; Zhang et al., 2023) maintain feature banks for pseudo-label refinement but introduce prohibitive computational overhead for memory indexing and require structural modifications incompatible with frozen VLM backbones. Recent works attempt to reduce overhead through sample selection (Niu et al., 2023; Tan et al., 2025), but these strategies focus on a small number of high-confidence samples, which induces catastrophic forgetting by overfitting and deviates from pretrained feature manifolds (Lee et al., 2024). Notably, our work reformulates test-time adaptation through uncertainty-weighted entropy minimization, which suppresses overconfidence on false positives while preserving frozen VLM backbones.

**Uncertainty in Cross-Modal Retrieval.** Uncertainty quantification has gained traction in cross-modal retrieval. Generally, uncertainty can be quantified as the discrepancy of representation between different modalities, which is more pronounced under domain gaps. (Chen et al., 2024) integrate fine- and coarse-grained retrieval with different fluctuations to model uncertainty and rectify the matching objective. Furthermore, (Li et al., 2024a) leverage subjective logic to select reliable cross-modal pairs and masked modeling to capture cross-modal relations, and also exploit multi-grained uncertainty-based alignments to mitigate domain shifts. With the help of an extra large vision-language model, (Zhao et al., 2024) use CLIP to reflect the uncertainty of input pairs and boost zero-shot performance via an uncertainty-aware reward feedback mechanism. (Li et al., 2025a) optimize the robustness of test-time adaptation via candidate selection, inter-modal gap learning, and intra-modal uniformity learning, yet are constrained to query modal shifts. Through a novel design of probabilistic distance metrics and hierarchical learning objectives, (Tang et al., 2025) explicitly model uncertainty at multi-grained levels, enabling more nuanced and robust composed image retrieval that can handle polysemy and ambiguity in search intentions. Recent cross-modal retrieval uncertainty estimation methods, whether multi-grained or contrastive, optimize representa-

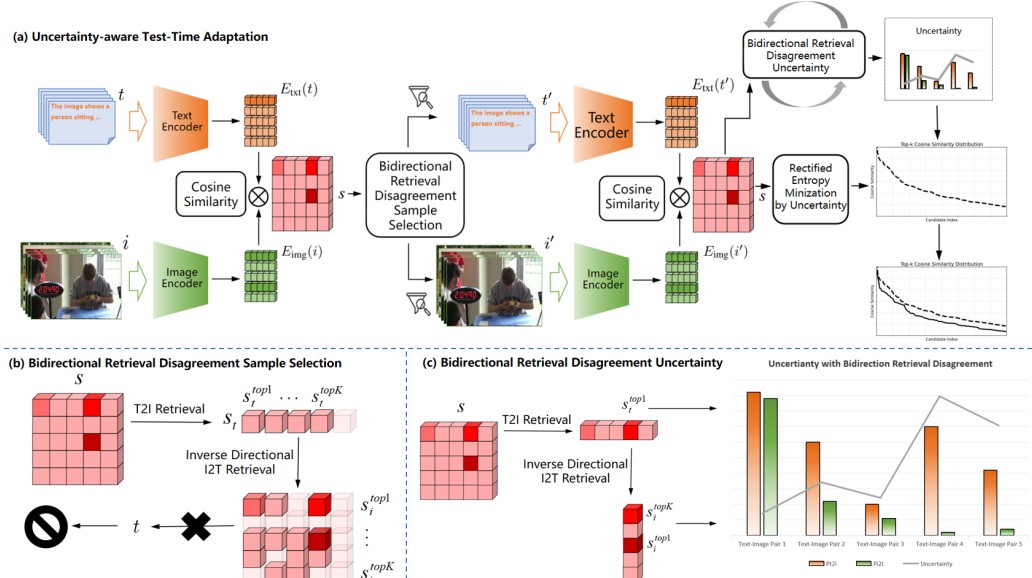

Figure 2: **Uncertainty-aware Test-Time Adaptation Framework(UATTA)**. As shown in (a), given the input image set $i$ and text set $t$ in the test set, we first extract the features respectively, and then calculate the cosine similarity $S$ for every pairs. As shown in (b), we sort the samples according to the similarity and select the image-text pairs who are mutual top-$K$ neighbors. After filtering out the noisy samples, based on the high-quality images $i'$ and texts $t'$, we further exploits the Bidirectional Retrieval Disagreement mechanism to estimate uncertainty with both text-to-image retrieval probability and inverse image-to-text probability, as detailed in (c). This uncertainty signal dynamically reweights the entropy minimization objective during test-time adaptation, suppressing overconfident updates on FP candidates and sharpening the similarity distribution toward true positives. Our UATTA framework mitigates domain gaps without computational overhead and any extra architecture.

tion similarity while neglecting retrieval trajectory consistency. Our bidirectional retrieval disagreement mechanism formulates an uncertainty score with the inherent retrieval trajectory-symmetric nature.

## 3 METHOD

In this section, we introduce the proposed Uncertainty-aware Test-Time Adaptation (UATTA) framework for text-based person search. As illustrated in Fig. 2, our method operates without access to labeled target-domain data during inference. First, we formalize the cross-modal retrieval task under both CLIP-based one-stage and XVLM-based two-stage paradigms, followed by a dynamic sample selection strategy based on the Bidirectional Retrieval Disagreement (BRD). Based on the filtered data, we present the uncertainty estimation. Finally, we integrate uncertainty into test-time adaptation via entropy recalibration.

**Problem Formulation.** For a given image $i$ and a given text $t$, the encoders of cross-modal retrieval model maps the image-text pair into a shared embedding space and output similarity scores. For CLIP-based one-stage framework, the similarity scores directly deploy the cosine similarity. For XVLM-based two-stage framework, an additional image-text matching (ITM) head is further applied to regress a matching score. Therefore, the similarity score of text-to-image retrieval can be derived as:

$$s(t,i) = \begin{cases} \cos\big(E_{\text{txt}}(t), E_{\text{img}}(i)\big), & \text{if one-stage retrieval model} \\ E_{\text{cross}}\big(E_{\text{txt}}(t), E_{\text{img}}(i)\big), & \text{if two-stage retrieval model} \end{cases} \quad (1)$$

where $\cos(\cdot,\cdot)$ is cosine similarity, and $E_{\text{txt}}$, $E_{\text{img}}$ are modality-specific encoders, and $E_{\text{cross}}$ denotes the cross-attention model from two-stage retrieval models.

**Bidirectional Retrieval Sample Selection.** For text-to-image retrieval, we define $\text{rank}_i(t)$ as the rank of image $i$ when retrieving using text $t$ to find image $i$, while $\text{rank}_t(i)$ is similar for image-to-text retrieval. During test-time adaptation, we select reliable samples at first. The reliable sample is defined as mutual top-$K$ ranking query text. For a certain query text $t$, $i$ images with $\text{rank}_i(t) < K$ are reserved, and then each image in these $K$ images is mutually used to find $K$ texts satisfying $\text{rank}_t(i) < K$ respectively. If the initial certain query text $t$ is one of the $K \cdot K$ mutual top-$K$ ranking texts, this query text $t$ is selected as a reliable sample. This retains samples with moderate uncertainty, useful for generalization, while discarding highly inconsistent pairs, which have high-uncertainty. In ablation study (Sec. 4a) we discuss more details about the selection of $K$ accompanying in a bidirectional retrieval disagreement uncertainty perspective.

**Bidirectional Retrieval Disagreement as Uncertainty Proxy.** Following the definition of uncertainty in (Kendall & Gal, 2017), we propose that the uncertainty of a retrieval model can be measured by observing its behavior. Specifically, we posit that the epistemic uncertainty of a retrieval model can be effectively quantified by measuring the disagreement between its bidirectional retrieval predictions. We define the model's uncertainty as the variance of its parameters, $\text{Unc}(\theta) := \text{Var}(\theta)$, a standard definition from a Bayesian perspective where parameters are treated as random variables. A large $\text{Var}(\theta)$ signifies high uncertainty in the learned weights. To measure this uncertainty in practice, we propose a proxy called **Bidirectional Retrieval Disagreement**, $D(t, i)$, defined as the difference between the text-to-image and image-to-text retrieval probabilities of the distinct text-to-image ($p_{t2i}$) and image-to-text ($p_{i2t}$) functions for a given pair $(t, i)$:

$$D(t, i) := ||p_{t2i}(y|t, i, \theta) - p_{i2t}(y|t, i, \theta)||. \tag{2}$$

where $y$ is the ground-truth label denoting wether $i$ and $t$ is truely-matched or not, and $p_{t2i}(y|t, i, \theta)$, $p_{i2t}(y|t, i, \theta)$ denote the probabilities of text-to-image and image-to-text search predictions. While the functions $p_{t2i}$ and $p_{i2t}$ depend on different subsets of parameters (*e.g.*, separate prediction heads), our analysis considers uncertainty over the entire parameter vector $\theta$. Our theoretical justification aims to prove the proportionality $\text{Var}(\theta) \propto D(t, i)$.

The proof is based on the principle of symmetric consistency. An idealized model with zero uncertainty ($\text{Var}(\theta) = 0$) can be represented by a single set of optimal parameters, $\theta_0 = E[\theta]$. Such a deterministic model, if well-trained, should exhibit symmetric predictions, meaning the probability of retrieving $i$ from $t$ is consistent with retrieving $t$ from $i$. Consequently, for this ideal model, the prediction disagreement is negligible:

$$D(t, i)|_{\theta=\theta_0} = ||p_{t2i}(y|t, i, \theta_0) - p_{i2t}(y|t, i, \theta_0)|| \approx 0. \tag{3}$$

In a realistic model, however, uncertainty implies that $\text{Var}(\theta) > 0$. The parameters $\theta$ are subject to perturbations around their mean $\theta_0$. These parameter perturbations disrupt the model's symmetric consistency, as they affect the distinct computational paths of $p_{t2i}$ and $p_{i2t}$ differently.

To formalize the relationship between parameter variance and prediction disagreement, we analyze the effect of these perturbations using a first-order Taylor expansion of the prediction functions around $\theta_0$ (Here we omit $t$ and $i$):

$$p_{t2i}(y|\theta) \approx p_{t2i}(y|\theta_0) + (\theta - \theta_0)^T \nabla_\theta p_{t2i}(y|\theta_0) \tag{4}$$

$$p_{i2t}(y|\theta) \approx p_{i2t}(y|\theta_0) + (\theta - \theta_0)^T \nabla_\theta p_{i2t}(y|\theta_0) \tag{5}$$

By substituting these into the definition of $D(t, i)$, we obtain:

$$D(t, i) \approx ||(p_{t2i}(y|\theta_0) - p_{i2t}(y|\theta_0)) + (\theta - \theta_0)^T (\nabla_\theta p_{t2i}(y|\theta_0) - \nabla_\theta p_{i2t}(y|\theta_0))||$$

Applying the symmetric consistency assumption, where $p_{t2i}(y|\theta_0) - p_{i2t}(y|\theta_0) \approx 0$, the expression simplifies to:

$$D(t, i) \approx ||(\theta - \theta_0)^T (\nabla_\theta p_{t2i}(y|\theta_0) - \nabla_\theta p_{i2t}(y|\theta_0))||. \tag{6}$$

This result demonstrates that the magnitude of the prediction disagreement $D(t, i)$ is directly dependent on the parameter deviation $(\theta - \theta_0)$. Since $\text{Var}(\theta) = E[(\theta - \theta_0)^2]$ measures the expected squared magnitude of this deviation, a larger parameter variance will lead to a larger expected prediction disagreement. This establishes the proportionality $\text{Var}(\theta) \propto D(t, i)$, validating the use of prediction disagreement as a computationally efficient and theoretically grounded proxy for model

uncertainty. In practice, we apply the softmax similarity as the probability, and the model parameter $\theta$ is in the similarity function $s$ as:

$$p_{t2i}(y|t_k, i_k) = \frac{\exp(s(t_k, i_k)/\tau)}{\sum_{j=1}^N \exp(s(t_k, i_j)/\tau)}, \quad p_{i2t}(y|t_k, i_k) = \frac{\exp(s(i_k, t_k)/\tau)}{\sum_{j=1}^N \exp(s(i_k, t_j)/\tau)} \quad (7)$$

**Uncertainty-aware Test-Time Adaptation (UATTA).** Standard test-time adaptation (TTA) minimizes entropy on test data, but often overfits to false positives. Therefore, we propose to harness BRD-based uncertainty to recalibrate gradients. In practise, we observe the mean probability value also reflect the matching degree, and thus we jointly take the mean value into consideration, modifying the $D(t, i)$ as:

$$D(t, i) := \exp\left(\frac{|p_{t2i}(y|t,i) - p_{i2t}(y|t,i)|}{\frac{p_{t2i}(y|t,i)+p_{i2t}(y|t,i)}{2}}\right). \quad (8)$$

Following the multiple classification formulation in (Wang et al., 2021) and Eq. 8, we define the UATTA objective of text-to-image retrieval as:

$$\mathcal{L}_{\text{UATTA}} = \sum_{(t,i)\in\mathcal{S}} \left(\frac{-\sum p_{t2i}(y|t,i)\log(p_{t2i}(y|t,i))}{D(t,i)} + \frac{-\sum p_{i2t}(y|t,i)\log(p_{i2t}(y|t,i))}{D(t,i)}\right), \quad (9)$$

where $\mathcal{S}$ is the filtered image-text pairs set after Bidirectional Retrieval Sample Selection. The weight $D(t, i)$ down-weights high-uncertainty pairs. This achieves Bidirectional Retrieval Disagreement calibration. For true positives, $D(t, i)$ is small, so gradients strongly pull $t$ and $i$ together. For ambiguous or false matches, $D(t, i)$ is large, suppressing misleading updates. Compared to TCR (Li et al., 2025a), which enforces uniformity and alignment via explicit feature-space constraints, UATTA implicitly optimizes the embedding space by leveraging the inherent consistency of correct retrieval. A query should not only retrieve its target image, but the corresponding image should also retrieve the query. This dual consistency naturally separates TP from FP without auxiliary objectives. Thus, UATTA enables effective, label-free adaptation under domain shift, with minimal computational overhead and architectural adjustment.

## 4 EXPERIMENT

**Experiment Setting.** We conduct experiments on two distinct frameworks for text-based person search: a one-stage retrieval framework and a two-stage retrieve-and-match framework. These choices allow us to evaluate our approach on tasks with varying complexity, from standard retrieval to fine-grained matching. **(1) CLIP-based One-Stage Framework.** For the standard person retrieval task, we adopt the state-of-the-art LuPerson-HAM model as our baseline. Our experiments are conducted on three real-world benchmarks: RSTPReid, CUHK-PEDES, and ICFG-PEDES. A key challenge is that LuPerson-HAM is pre-trained on synthetic annotations, which creates a significant domain gap compared to the human-annotated captions in the test sets. Our test-time adaptation method is designed to bridge this gap. **(2) XVLM-based Two-Stage Framework.** For the more complex person anomaly search task, which requires both coarse-grained retrieval and fine-grained matching, we follow the state-of-the-art CMP model. This model, based on the XVLM architecture, is evaluated on the PAB benchmark. Similar to the one-stage setup, PAB's training data is synthetically generated, while its test data consists of real-world images with human-corrected captions, presenting a clear domain gap that motivates our approach.

**Implementation Details.** During test-time adaptation, we optimize only the affine parameters ($\gamma$ and $\beta$) of the Layer Normalization layers within the final six layers of the text encoder. This specific choice is made to maintain consistency with the CMP baseline, where these last six layers correspond to the cross-modal attention blocks essential for image-text matching. We adopt the AdamW optimizer for all experiments. For the LuPerson-HAM baseline, the learning rate is set to $1e-3$, with a batch size of 32 and a positive-to-negative sample ratio of 1:3. For the XVLM baseline, the learning rate is $1e-4$, the batch size is 16, and the sample ratio is 1:7. The number of adaptation rounds is adjusted based on the test set size, *i.e.*, 50 for PAB and RSTP-Reid, and 10 for ICFG-PEDES and CUHK-PEDES.

Table 1: Quantitative comparison of our proposed **Pretrain-then-Adapt** paradigm with state-of-the-art methods on on Text-based Person Anomaly Search benchmark PAB (Yang et al., 2024b). The gpu used for post-training is NVIDIA GeForce RTX 3090 GPU. Best results are **bold**. Second best results are underline.

**Person Anomaly Search Benchmark**

| Method | Training Type | Post-training Time | R@1 | R@5 | R@10 | mAP |
|---|---|---|---|---|---|---|
| MRA (Yang et al., 2025) | Pretrain | - | 9.91 | 23.66 | 31.45 | 17.15 |
| RaSa (Bai et al., 2023a) | Pretrain | - | 21.74 | 27.30 | 27.96 | 24.35 |
| WoRA (Sun et al., 2025) | Pretrain | - | 22.25 | 45.91 | 53.54 | 33.39 |
| APTM (Yang et al., 2023) | Pretrain | - | 22.90 | 45.80 | 52.38 | 33.56 |
| CAMeL (Yu et al., 2025) | Pretrain | - | 24.47 | 50.00 | 58.75 | 36.75 |
| IRRA (Jiang & Ye, 2023) | Pretrain | - | 30.59 | 59.61 | 68.91 | 44.41 |
| CLIP (Radford et al., 2021) | Pretrain | - | 47.57 | 81.55 | 89.03 | 62.73 |
| X-VLM (Zeng et al., 2021) | Pretrain | - | 71.94 | 97.78 | 98.99 | 83.96 |
| MRA (Yang et al., 2025) | Pretrain-then-Finetune | 1.06 hours on 4 gpus | 70.53 | 94.69 | 97.47 | 81.59 |
| APTM (Yang et al., 2023) | Pretrain-then-Finetune | 0.51 hours on 4 gpus | 72.14 | 95.30 | 97.17 | 82.78 |
| CAMeL (Yu et al., 2025) | Pretrain-then-Finetune | 1.01 hours on 4 gpus | 74.30 | 96.79 | 98.84 | 84.20 |
| WoRA (Sun et al., 2025) | Pretrain-then-Finetune | 0.88 hours on 4 gpus | 74.47 | 96.82 | 98.48 | 84.60 |
| IRRA (Jiang & Ye, 2023) | Pretrain-then-Finetune | 19.6 hours on 4 gpus | 76.39 | 97.62 | 99.14 | 86.33 |
| CLIP (Radford et al., 2021) | Pretrain-then-Finetune | 18.4 hours on 4 gpus | 77.60 | 98.84 | 99.75 | 87.35 |
| RaSa (Bai et al., 2023a) | Pretrain-then-Finetune | 0.74 hours on 4 gpus | 80.79 | 98.89 | 99.65 | 89.20 |
| X-VLM (Zeng et al., 2021) | Pretrain-then-Finetune | 40.5 hours on 4 gpus | 81.95 | 98.84 | 99.19 | 89.86 |
| X-VLM + CMP (Yang et al., 2024b) | Pretrain-then-Finetune | 48.1 hours on 4 gpus | 84.93 | 99.09 | 99.75 | 91.66 |
| X-VLM + SAR (Niu et al., 2023) | Pretrain-then-Adapt | 0.38 hours on 1 gpu | 73.20 | 97.87 | 99.09 | 84.58 |
| X-VLM + Tent (Wang et al., 2021) | Pretrain-then-Adapt | 0.23 hours on 1 gpu | 73.50 | 95.65 | 97.57 | 83.71 |
| X-VLM + SHOT (Liang et al., 2020) | Pretrain-then-Adapt | 0.26 hours on 1 gpu | 73.66 | 95.80 | 97.82 | 83.97 |
| X-VLM + READ (Yang et al., 2024a) | Pretrain-then-Adapt | 0.23 hours on 1 gpu | 74.62 | 96.00 | 98.18 | 84.61 |
| X-VLM + TCR (Li et al., 2025a) | Pretrain-then-Adapt | 0.25 hours on 1 gpu | 74.92 | 96.15 | 97.97 | 84.72 |
| **X-VLM + Ours** | Pretrain-then-Adapt | **0.08 hours on 1 gpu** | **76.13** | **98.02** | **99.09** | **86.14** |

## 4.1 COMPARISON WITH STATE-OF-THE-ARTS

**Comparison with Pretrain Models.** We compare our method with state-of-the-art methods on multiple benchmarks. As shown in Table 1, our method significantly improves +4.19% R@1 compared to pretrained XVLM, which proves the capacity of our Pretrain-then-Adapt paradigm on mitigating domain gaps between unrelated pretrained data and specific person anomaly search data. Notably, our pretrain-then-adapt paradigm achieves significant efficiency gains with merely 0.08 hours of adaptation time. The process of adaptation operates directly on unlabeled test data of target domain, while others need finetuning on labeled train data of target domain, consuming additional computational burden. Although some models, benefiting from lightweight fine-tuning modules, reduce post-training time from dozens of hours to approximately one hour, they still require 4 NVIDIA GeForce RTX 3090 GPU whereas only single 3090 GPU for ours. The efficiency gains become particularly significant when considering practical deployment constraints in privacy-sensitive adn resource-constrained environments. We observe a similar improvement on three text-based person search benchmarks in Table 2. The results show that the R@1 score increases 2.35%, 0.33% and 1.51% on RSTPReid, CUHK-PEDES and ICFG-PEDES respectively, and the mAP score is improved by 2.26, 0.29 and 0.57. These boostes underscore the efficacy of our proposed bidirectional retrieval disagreement uncertainty and sample selection in mitigating the impact from false positives, which generally refines model to be overconfident in traditional entropy minimization test-time adaptation methods.

**Comparison with other TTA methods.** We modify others test-time adaptation methods i.e.Tent(Wang et al., 2021), SHOT(Liang et al., 2020), SAR(Niu et al., 2023), READ(Yang et al., 2024a), TCR(Li et al., 2025a) from fully Test-Time Adaptation paradigm(Wang et al., 2021) to our Pretrain-then-Adapt paradigm on RSTPReid, CUHK-PEDES and ICFG-PEDES. As shown in Table 1, Our method demonstrates superior performance and efficiency, achieving gains of 1.21% in

Table 2: Quantitative comparison of our proposed **Pretrain-then-Adapt** paradigm with state-of-the-art direct transfer models and other existing **Test-Time-Adaptation** and **Semi/Un-supervised** methods on real-world text-based person search benchmarks (Li et al., 2017; Zhu et al., 2021; Ding et al., 2021). Best results are **bold**. Second best results are underline.

**Text-based Person Search Benchmarks**

| Method | RSTPReid | | | | CUHK-PEDES | | | | ICFG-PEDES | | | |
|---|---|---|---|---|---|---|---|---|---|---|---|---|
| | R1 | R5 | R10 | mAP | R1 | R5 | R10 | mAP | R1 | R5 | R10 | mAP |
| Pretrain | | | | | | | | | | | | |
| CLIP (Radford et al., 2021) | 12.65 | 27.16 | – | 11.15 | 6.67 | 17.91 | – | 2.51 | 13.45 | 33.85 | – | 10.31 |
| LuPerson-T (Shao et al., 2023) | 22.40 | – | – | 17.08 | 21.88 | – | – | 19.96 | 11.46 | – | – | 4.56 |
| SYNTH-PEDES (Zuo et al., 2024) | 42.69 | – | – | 31.18 | 57.58 | – | – | 52.45 | 57.08 | – | – | 32.06 |
| LuPerson-MLLM (Tan et al., 2024) | 51.65 | 74.20 | 82.85 | 38.31 | 38.29 | 56.60 | 64.56 | 20.43 | 57.61 | 75.99 | 82.76 | 51.45 |
| LuPerson-HAM (Jiang et al., 2025) | 59.50 | 80.05 | 87.05 | 44.11 | 70.59 | 86.89 | 91.78 | 63.39 | 60.64 | **77.50** | **83.26** | 35.54 |
| Unsupervised | | | | | | | | | | | | |
| GAAP (Li et al., 2024b) | 44.45 | 65.15 | 75.30 | 31.21 | 47.64 | 67.79 | 76.08 | 41.28 | 27.12 | 44.91 | 53.56 | 11.43 |
| GTR (Bai et al., 2023b) | 46.65 | 70.70 | 80.65 | 34.95 | 48.49 | 68.88 | 76.51 | 43.67 | 29.64 | 47.23 | 55.54 | 14.20 |
| PSPD (Chen et al., 2025) | 48.50 | 69.95 | 78.50 | 34.83 | 53.47 | 72.81 | 76.57 | 46.41 | 38.49 | 53.40 | 60.35 | 16.49 |
| MUMA (Li et al., 2025b) | 54.35 | 76.05 | 83.65 | 40.50 | 59.52 | 77.79 | 84.65 | 52.75 | 38.11 | 56.01 | 63.96 | 19.02 |
| Semi-supervised | | | | | | | | | | | | |
| CMMT (Zhao et al., 2021) | – | – | – | – | 57.10 | 78.14 | 85.23 | – | – | – | – | – |
| Generation-then-Retrieval (Gao et al., 2025) | 56.45 | – | – | 44.45 | 63.87 | – | – | 57.18 | 46.46 | – | – | 26.90 |
| TextReID (Han et al., 2021) | – | – | – | – | 64.40 | 81.27 | 87.96 | 61.19 | – | – | – | – |
| ECCA (Gong et al., 2024) | – | – | – | – | 68.13 | **87.26** | 91.88 | - | – | – | – | – |
| Pretrain-then-Adapt | | | | | | | | | | | | |
| LuPerson-HAM + CoOp (Zhou et al., 2022) | 58.60 | 79.65 | 87.50 | 43.65 | 70.09 | 86.48 | 91.32 | 63.10 | 60.28 | 76.24 | 82.31 | 35.16 |
| LuPerson-HAM + SAR (Niu et al., 2023) | 59.55 | 80.05 | 87.00 | 44.12 | 70.63 | 86.87 | 91.79 | 63.40 | 60.64 | 77.50 | 83.25 | 35.54 |
| LuPerson-HAM + Tent (Wang et al., 2021) | 59.65 | 79.75 | 87.30 | 44.24 | 70.30 | 87.02 | 91.74 | 63.26 | 59.59 | 76.89 | 82.85 | 34.86 |
| LuPerson-HAM + READ (Yang et al., 2024a) | 59.80 | 79.90 | 87.30 | 44.37 | 70.06 | 86.98 | 91.82 | 63.12 | 60.31 | 77.09 | 82.96 | 35.27 |
| LuPerson-HAM + SHOT (Liang et al., 2020) | 60.10 | 79.85 | 87.10 | 44.46 | 70.43 | 86.90 | 91.99 | 63.30 | 60.31 | 76.95 | 82.86 | 35.10 |
| LuPerson-HAM + TCR (Li et al., 2025a) | 61.00 | 80.85 | 88.35 | 45.94 | 70.66 | 87.21 | **92.13** | **63.60** | 59.32 | 75.63 | 81.63 | 35.13 |
| LuPerson-HAM + **Ours** | **61.85** | **81.40** | **88.40** | **46.37** | **70.92** | 86.89 | 91.86 | 63.50 | **62.15** | 77.31 | 82.95 | **36.11** |

R@1 and 1.42% in mAP over all compared baselines, with 0.15 fewer hours of adaptation time on 1 gpu. As shown in Table 2, it is obviously on ICFG-PEDES that all test-time adaptation methods fail and our method outperforms baseline by 1.51% R@1 and 0.57% mAP, but our method also has performance degradation at R@5 and R@10, because the bidirectional retrieval disagreement mechanism is designed to rectify the harmfulness from top-1 false positives and neglects possible potential true positives in top-2 to top-10 range. This is a future direction for us to explore smooth utilization of these potential true positives in edge zone. Similar situation occurs on CUHK-PEDES, as our method achieves best R@1 of 70.92% but is inferior to TCR on R@5, R@10 and mAP. On RSTPReid, our method surpasses other existing methods.

**Comparison with other Semi/Un-supervised Methods.** Generally, unsupervised (Li et al., 2025b; Chen et al., 2025; Bai et al., 2023b; Li et al., 2024b) and semi-supervised (Gao et al., 2025; Han et al., 2021; Zhao et al., 2021; Gong et al., 2024) paradigms for text-based person search leverage advanced VLMs to synthesize pseudo-annotations, serving as proxies for supervised image-text pairs. However, this reliance on synthetic data inevitably introduces intrinsic domain shifts. In contrast, our approach performs direct adaptation on the test data. Despite the absence of ground-truth pairings, the textual descriptions remain aligned with the target domain. Consequently, our method focuses on mitigating the distribution shifts of the pretrained model, avoiding the noisy discriminative supervision characteristic of prior approaches. As evidenced in Table 2, existing unsupervised and semi-supervised methods struggle to fully leverage the pretrained model's capacity, often compromising representation quality due to label noise.

Table 3: Comparison of **Uncertainty Formulations** on RSTPReid (Zhu et al., 2021) benchmark. The 'baseline' refers to directly using the CLIP (Radford et al., 2021) encoders pretrained by LuPerson-HAM (Jiang et al., 2025). $p_{t2i}$ is the text-to-image retrieval probability, $p_{i2t}$ is the inverse retrieval probability, which uses the gallery image from $p_{t2i}$ to retrieve the corresponding query text. $N_{t2i}$ is the size of image gallery per identity. $N_{i2t}$ is the size of text query per identity. $\epsilon$ is a small constant to prevent divided by zero. $\mathbf{s}_{t2i}^{\text{top-}K}$ denotes the top-$K$ similarity matrix of text-to-image retrieval. Equally, $\mathbf{s}_{i2t}^{\text{top-}K}$ denotes the top-$K$ similarity matrix of inverse directional image-to-text retrieval. The similarity matrix is transformed by a softmax function to obtain retrieval probabilities for the top-$K$ results. Best results are **bolded**.

| Index | Uncertainty Formulation | Bidirectional Retrieval Probability | RSTPReid | | | |
| --- | --- | --- | --- | --- | --- | --- |
| | | | R1 | R5 | R10 | mAP |
| 1 | $\exp\left(1 - \dfrac{p_{t2i} + p_{i2t}}{2}\right)$ | $p_{t2i} = \text{softmax}(\mathbf{s}_{t2i}^{\text{top-}K})$ $p_{i2t} = \text{softmax}(\mathbf{s}_{i2t}^{\text{top-}K})$ | 61.40 | 80.50 | 87.75 | 46.16 |
| 2 | $\|\log(p_{t2i} + \varepsilon) - \log(p_{i2t} + \varepsilon)\|$ | $p_{t2i} = \text{softmax}(\mathbf{s}_{t2i}^{\text{top-}K})$ $p_{i2t} = \text{softmax}(\mathbf{s}_{i2t}^{\text{top-}K})$ | 61.75 | 81.30 | 88.40 | 45.88 |
| 3 | $\exp\left(\dfrac{\|p_{t2i} - p_{i2t}\|}{\frac{p_{t2i} + p_{i2t}}{2}}\right)$ | $p_{t2i} = \text{softmax}(\mathbf{s}_{t2i}^{\text{top-}K})$ $p_{i2t} = \text{softmax}(\mathbf{s}_{i2t}^{\text{top-}K})$ | **61.85** | 81.40 | 88.40 | 46.37 |
| 4 | $\exp\left(\dfrac{\|p_{t2i} \cdot N_{t2i} - p_{i2t} \cdot N_{i2t}\|}{\frac{p_{t2i} \cdot N_{t2i} + p_{i2t} \cdot N_{i2t}}{2}}\right)$ | $p_{t2i} = \text{softmax}(\mathbf{s}_{t2i}^{\text{top-}K})$ $p_{i2t} = \text{softmax}(\mathbf{s}_{i2t}^{\text{top-}K})$ | 61.75 | 81.35 | 88.60 | **46.58** |
| 5 | $\exp\left(\dfrac{\|p_{t2i} - p_{i2t} \cdot \frac{N_{i2t}}{N_{t2i}}\|}{\frac{p_{t2i} + p_{i2t} \cdot \frac{N_{i2t}}{N_{t2i}}}{2}}\right)$ | $p_{t2i} = \text{softmax}(\mathbf{s}_{t2i}^{\text{top-}K})$ $p_{i2t} = \text{softmax}(\mathbf{s}_{i2t}^{\text{top-}K})$ | 61.70 | 81.45 | 88.60 | 46.57 |
| 6 | $\exp\left(\dfrac{\|p_{t2i} \cdot N_{t2i} - p_{i2t} \cdot N_{i2t}\|}{\frac{p_{t2i} \cdot N_{t2i} + p_{i2t} \cdot N_{i2t}}{2}}\right)$ | $p_{t2i} = \text{softmax}(\mathbf{s}_{t2i}^{\text{top-}K} \cdot N_{t2i})$ $p_{i2t} = \text{softmax}(\mathbf{s}_{i2t}^{\text{top-}K} \cdot N_{i2t})$ | 61.75 | **81.90** | **88.90** | 46.47 |
| 7 | $\exp\left(\dfrac{\|p_{t2i}/N_{t2i} - p_{i2t}/N_{i2t}\|}{\frac{p_{t2i}/N_{t2i} + p_{i2t}/N_{i2t}}{2}}\right)$ | $p_{t2i} = \text{softmax}(\mathbf{s}_{t2i}^{\text{top-}K} \cdot N_{t2i})$ $p_{i2t} = \text{softmax}(\mathbf{s}_{i2t}^{\text{top-}K} \cdot N_{i2t})$ | 61.30 | 81.40 | 88.55 | 46.04 |

## 4.2 ABLATION STUDIES AND FURTHER DISCUSSION

**Effect of Uncertainty Formulation.** We present an ablation study on the formulation of uncertainty in Table 3. The core thought is to assign a lower uncertainty on both top retrieval directions and a higher uncertainty while only uni-directional retrieval works. Formulation 1 in Table 3 only considers the higher similarity of true positives but ignore the difference between TP and FP. At a opposite perspective, formulation 2 focuses on the difference neglecting the absolute numerical magnitude. Combining with two views, formulation 3 achieves best score on RSTPReid, while the others are scaled version based on formulation 3 to balance the number of positive samples in the two retrieval directions. The extreme amplifications and balances destroy the suitable consistent distribution of TP and FP, then consequently weaken performance at R@1 score, which is the primary standard we use to choose formulation.

**Effect of $K$ Mutual Neighbours.** We conduct an ablation study of the choice of hyper-parameter $K$ in bidirectional retrieval disagreement sample selection in Table 4a. We choose $K$=5 as the default setting, due to the physical meanings, number of ground-truth labels. In specific, $K$=5 reserves the true positives with low uncertainty and enhanced generalization ability by involving image-text pairs with moderate uncertainty, which allows the uncertainty play a proper role in adjusting various pairs and avoid uncertainty invalid. Then, $K$=1 means that the adaptation process is optimized only by image-text pairs with low uncertainty and lacks generalization. While $K$=∞ denotes using all text-image pairs without any sample selection, false positives with high uncertainty appear to hurt performance.

**Effect of Negative Samples.** In Table 4b, we compare two experiments of the ratio between positive and negatives for one query. The optimal performance is presented with configuration of 1 : 3 on

RSTPReid. This suggests that suitable choice of ratio enhances the adaptation process with softmax entropy based on multiple classification.

Table 4: Ablation Study on RSTPReid benchmarks. (a) **Ablation of Bidirectional Top-$K$ Retrieval Consistent Sample Selection $K$.** $K$ denotes the mutual top range of bidirectional retrieval. As the number of ground-truth image in RSTPReid is 5, which represents the borderline of true and false positives, we choose $K$=5 with gray line as default setting. $K$=5 ensures the generalization of selected samples and filters out the extreme false positives. (b) **Ablation of the ratio between positive and negatives.** We apply different ratio of positive and negatives to compute entropy. The ratio of 1 : 3 improves the stability in test-time adaptation. Our default setting is in gray .

(a)

| $K$ | R@1 | R@5 | R@10 | mAP |
|---|---|---|---|---|
| 1 | 61.60 | 81.10 | 88.75 | 45.90 |
| 5 | 61.85 | 81.40 | 88.40 | 46.37 |
| $\infty$ | 61.55 | 80.75 | 88.20 | 45.96 |

(b)

| Ratio | R@1 | R@5 | R@10 | mAP |
|---|---|---|---|---|
| 1 : 3 | **61.85** | **81.40** | **88.40** | **46.37** |
| 1 : 7 | 61.55 | 81.35 | 88.75 | 46.32 |

## 5 CONCLUSION

In this work, we introduce a practical and label-free Pretrain-then-Adapt framework, challenging the conventional Pretrain-then-Finetune paradigm in text-based person search. To this end, we propose Uncertainty-Aware Test-Time Adaptation (UATTA), a novel method that leverages unlabeled test data to dynamically recalibrate model predictions under domain shift. Central to UATTA is a theoretically motivated Bidirectional Retrieval Disagreement (BRD) mechanism, which estimates uncertainty through the discrepancy between text-to-image and image-to-text retrieval probabilities. This uncertainty signal effectively identifies and suppresses overconfident false positives while preserving alignment for true positives. Extensive experiments across four benchmarks and two representative architectures, including CLIP-based one-stage and XVLM-based two-stage models, demonstrate that UATTA consistently improves retrieval performance without access to target-domain annotations or architectural modifications. Ablation studies validate the design of uncertainty formulation, highlighting the importance of mutual top-$K$ neighbors for balancing reliability and generalization. While our method focuses on retrieval consistency, future work will explore extending BRD to other cross-modal tasks such as image captioning or visual grounding. We hope UATTA inspires a shift toward uncertainty-conscious, deployment-friendly adaptation strategies in real-world vision-language systems.

## ETHICS STATEMENT

We hereby solemnly declare that **we have carefully read the ICLR Code of Ethics, and that this research strictly adheres to these guidelines**.

This work is motivated by the important goal of enhancing public safety and security through improved person retrieval systems. Our research on text-based person search has significant potential for positive societal impact in legitimate applications such as:

- **Public Safety Enhancement**: Assisting law enforcement in locating missing persons and identifying suspects through textual descriptions
- **Security Applications**: Improving surveillance systems for crime prevention and investigation in public spaces
- **Emergency Response**: Enhancing search and rescue operations by enabling more accurate person identification from witness descriptions

We have conducted this research with a strong commitment to ethical principles, including privacy protection, fairness, and transparency. The datasets used in this study are publicly available benchmarks specifically designed for academic research in person retrieval. We believe that advancing the state-of-the-art in text-based person search technology, when developed and deployed responsibly, can contribute significantly to public welfare and community safety. Our work is intended solely for legitimate, beneficial applications that respect individual rights and promote social good.

## REPRODUCIBILITY STATEMENT

We are committed to ensuring the reproducibility of this work. To ensure the reproducibility of our research, we will provide comprehensive code, data, and experimental details.

## LARGE LANGUAGE MODEL USAGE STATEMENT

In accordance with policy on Large Language Model (LLM) usage, we provide this statement to transparently disclose the extent of LLM involvement in our research.

**Usage Declaration:** LLMs were used exclusively for minor text polishing and grammatical refinement in the final stages of manuscript preparation. Specifically, we employed GPT-4 for:

- Correcting grammatical errors and improving sentence fluency
- Ensuring consistent academic tone throughout the paper
- Minor rephrasing of complex sentences for better readability

**Important Limitations:**

- **No conceptual contribution:** LLMs were not involved in research ideation, experimental design, or methodological development
- **No technical content generation:** All mathematical formulations, algorithms, and technical insights originated from the human authors
- **No data analysis:** LLMs played no role in data processing, experimental results, or statistical analysis
- **Human oversight:** All LLM-suggested edits were carefully reviewed and validated by the authors

The core intellectual contributions, formulation designs, and all experimental work, remain entirely the product of human authors. The LLM served purely as an auxiliary tool for language refinement, analogous to traditional proofreading services.

We confirm that this limited usage does not constitute significant LLM contribution that would warrant co-authorship or affect the original intellectual property of this work.

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

# A  ADDITIONAL ABLATION STUDIES

## A.1  COMPARISON WITH LIGHTWEIGHT TUNING METHODS.

Table A.1: Comparison of lightweight tuning methods on RSTPReid(Zhu et al., 2021) benchmark.

| Method | Tuning Layers | R@1 | R@5 | R@10 | mAP |
|---|---|---|---|---|---|
| Baseline | – | 60.64 | 77.50 | 83.26 | 35.54 |
| CoOp*(Zhou et al., 2022) | Prompt Embedding | 58.60 | 79.65 | 87.50 | 43.65 |
| Prefix-Tuning*(Li & Liang, 2021) | Prefix Embedding | 26.25 | 51.45 | 63.75 | 23.14 |
| LoRA*(Hu et al., 2022) | LoRA Matrix | 49.80 | 73.80 | 82.70 | 38.61 |
| Ours | Normalization Layer | 61.85 | 81.40 | 88.40 | 46.37 |

We compare our method with lightweight tuning methods in Table A.1. Baseline is LuPerson-HAM (Jiang et al., 2025) and * means that we try different hyperparameters *i.e.* learning rate, number of virtual tokens, rank of LoRA(Hu et al., 2022) etc., for lightweight tuning methods and selected the best result. CoOp(Zhou et al., 2022), which is a prompt learning method and belong to few-shot learning, fails with adaptation objective of entropy minimization. It can be interpreted that additional prompt token embeddings need labeled data to mimic natural language feature in the token embedding space, instead of adjusting the cross-modal feature distribution in latent space. Additionally, Parameter Efficient Fine-Tuning (PEFT)(Han et al., 2024) provides a practical solution by efficiently adjusting the large models over the various downstream tasks. We also evaluated two representative PEFT methods, *i.e.*, Prefix-Tuning (Li & Liang, 2021) and LoRA (Hu et al., 2022), for test-time adaptation on the RSTPReid benchmark, however, these approaches proved ineffective in our experiments. Although the trainable parameters in PEFT are lightweight, Entropy Minimization fails to provide sufficient supervision for learning discriminative representations.

## A.2  MORE ABLATION STUDY ON RSTPREID.

Table A.2: **More Ablation of Bidirectional Top-$K$ Retrieval Consistent Sample Selection $K$ on RSTPReid.** $K$ denotes the mutual top range of bidirectional retrieval. We obtain the best result at $K = 3$, but the number of ground-truth image in RSTPReid is 5, as it represent the borderline of true and false positives. We choose $K = 5$ with gray line as default setting.

Table A.3

| $K$ | R@1 | R@5 | R@10 | mAP |
|---|---|---|---|---|
| 1 | 61.60 | 81.10 | **88.75** | 45.90 |
| 2 | 60.90 | 81.30 | **88.75** | 46.14 |
| 3 | **61.90** | 81.20 | 88.05 | **46.39** |
| 4 | 61.85 | 81.30 | 88.15 | 46.38 |
| 5 | 61.85 | **81.40** | 88.40 | 46.37 |
| 6 | 61.05 | 80.95 | 88.30 | 46.17 |
| 8 | 61.30 | 81.10 | 87.85 | 46.04 |
| 10 | 61.05 | 80.80 | 88.00 | 45.93 |
| $\infty$ | 61.55 | 80.75 | 88.20 | 45.96 |

**Effect of $K$ Mutual Neighbours.** Despite the physical meaning of top-$K$, we compare more $K$ in Table A.3. To some extent, results indicate the generalization performance of different $K$.

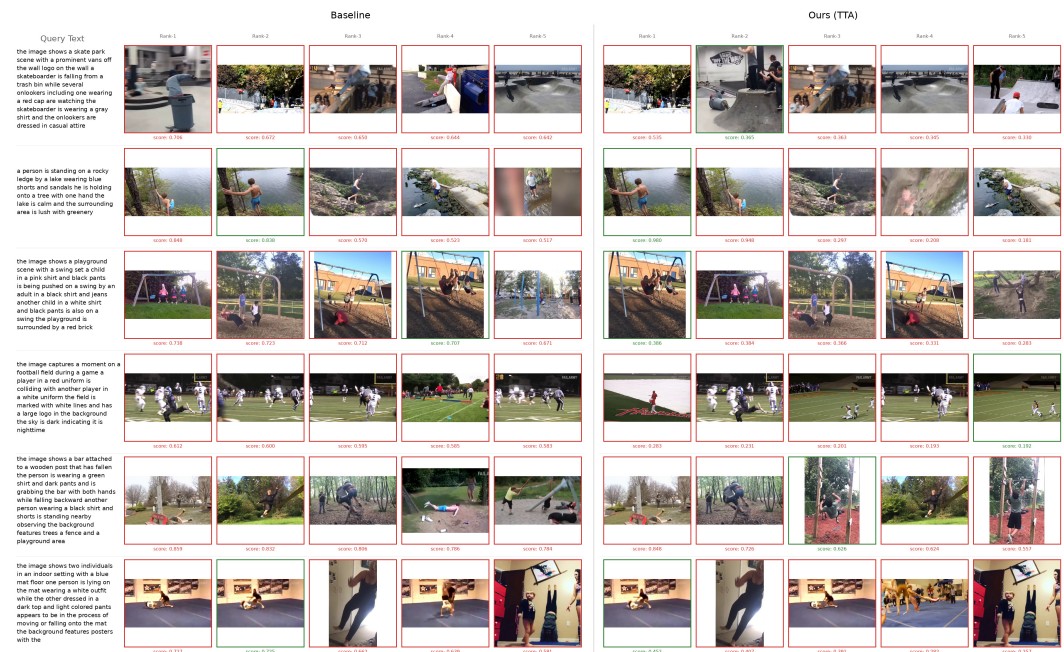

Figure B.1: **Top-**5 **Text-based Person Search Results on PAB.** The figure displays the Top-5 retrieval results for six representative text queries, with the confidence score for each rank provided below the corresponding image. The matched person images are annotated in green boxes, and the false ones are in red.

# B MORE QUALITATIVE RESULTS

## B.1 QUALITATIVE ANALYSIS OF PERSON SEARCH PERFORMANCE

To qualitatively validate the effectiveness of our Uncertainty-Aware Test-Time Adaptation (UATTA), we present a visual comparison of retrieval results between the Baseline and UATTA on the PAB dataset in Figure B.1. The visualization effectively showcases two key strengths of UATTA: First, in challenging cases (Rows 1, 4, 5) where the Baseline fails due to overly high confidence in false positives, UATTA successfully rectifies the score distribution by mitigating this over-confidence, leading to the correct identification of the ground-truth image. Second, for scenarios requiring fine-grained semantic distinction (Rows 2, 3, 6), UATTA leverages the bidirectional retrieval disagreement proxy to effectively disambiguate subtle differences between the text and image modalities. This mechanism allows UATTA to elevate the correct match from a low rank to Rank-1, significantly outperforming the Baseline. Overall, the qualitative results confirm that UATTA achieves a sharper, more robust, and accurate confidence distribution, validating its superiority in handling both retrieval ambiguity and fine-grained visual differences.

## B.2 VISUALIZATION OF FEATURE SPACE SHIFTS

In Figure B.2, T-SNE visualization provides an intuitive illustration of the impact of Test-Time Adaptation (TTA) on Feature Space. The visualization is focused on a representative subset of the Top-15 most frequent person identities to ensure clarity and showcase the adaptation effects vividly. We notice that the initial spread of original Query features (circles) demonstrates the significant domain gap and feature ambiguity present before adaptation, justifying the necessity of TTA. After TTA, regions circled by dotted ellipses indicate that query features, post-TTA (diamonds), are effectively adapted to align more closely with their respective gallery feature (squares) clusters. This convergence demonstrates the efficacy of TTA in reducing feature disparity and enhancing matching performance. While the majority of person identities show strong alignment, we observe that

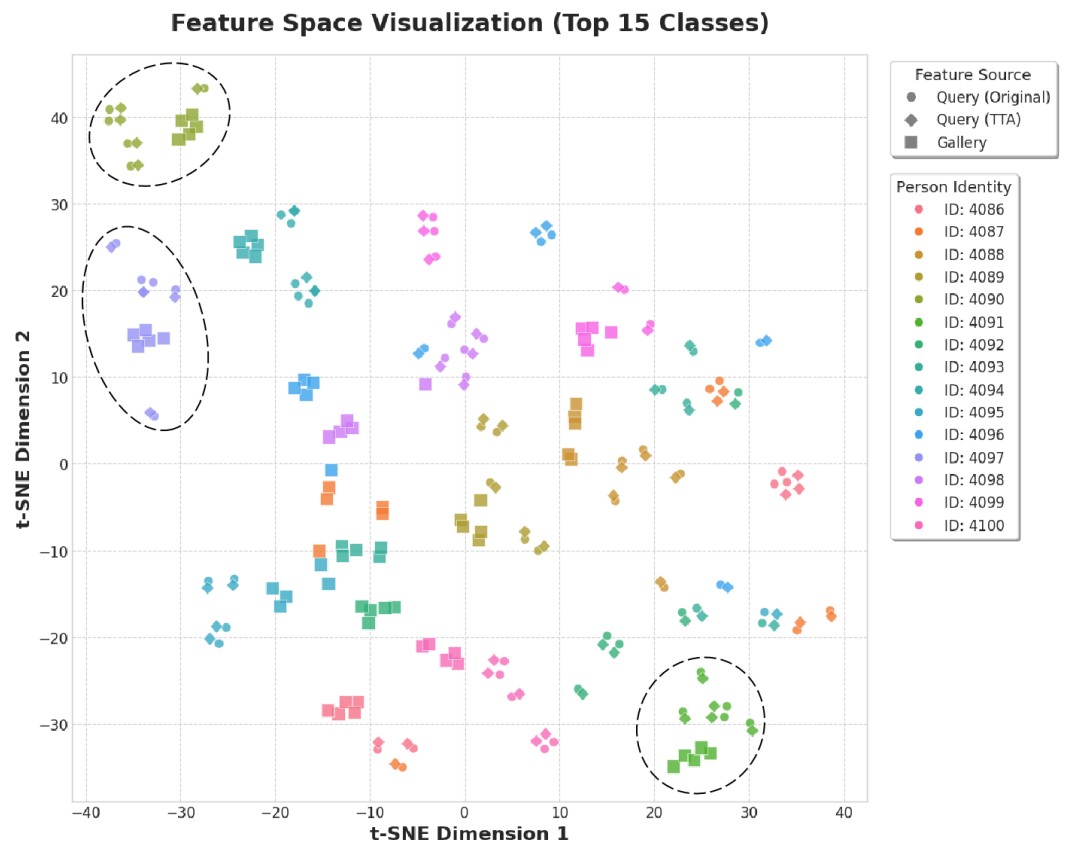

Figure B.2: **T-SNE Visualization of Feature Space Shifts on RSTPReid.** The three distinct point types represent: original query features (circles) before Test-Time Adaptation (TTA), query features after TTA (diamonds), and gallery features (squares). Different colors distinguish individual person identities.

some identities still exhibit residual ambiguity after TTA, suggesting potential avenues for future improvement in feature consolidation.

