# OpenReview forum: "Pretrain then Adapt: Uncertainty-Aware Test-Time Adaptation for Text-based Person Search"
_ICLR.cc/2026/Conference — Submitted to ICLR 2026_

### Official Review · Reviewer_aw8W · 2025-10-28

**Soundness:** 2
**Presentation:** 3
**Contribution:** 2
**Rating:** 2
**Confidence:** 4

**Summary:**

This paper pioneers the exploration of the text-based person retrieval task in the absence of labeled target-domain data. The authors propose an uncertainty-aware test-time adaptation framework based on a bidirectional retrieval disagreement mechanism, which measures the discrepancy between image-to-text and text-to-image matching. Extensive experiments demonstrate the effectiveness of the proposed method.

**Strengths:**

1. The paper is well written, and I am able to clearly understand most of its contributions.
2. This work pioneers the exploration of text-based person retrieval without relying on labeled target-domain data, which could alleviate the dependence on expensive human annotations.

**Weaknesses:**

1. **Logical Flaw**: The proposed bidirectional retrieval pipeline assumes access to the entire set of test textual queries in advance. In real-world scenarios, however, text-based person retrieval is typically performed in an online manner, where user queries arrive sequentially and are not known beforehand. This assumption significantly limits the practical applicability of the method, as it does not align with the dynamic nature of actual deployment environments.
2. **There is inconsistency in terminology**: the title refers to "text-based person search," whereas the main text uses "text-based person retrieval." Although both terms are acceptable, I recommend maintaining consistency throughout the paper.
3. **The paper lacks a unique and in-depth discussion of the text-based person retrieval task itself.** In fact, many existing vision-language cross-modal tasks (e.g., image/video-text retrieval) face similar challenges with expensive text annotation. Is the proposed method applicable to these tasks as well?
4. **The paper does not thoroughly discuss existing test-time adaptation methods.** While related work is mentioned, it remains unclear whether current approaches can be directly applied to text-based person retrieval, and what specific issues the proposed method addresses when adapting test-time adaptation techniques to this task.
5. **The experimental section lacks citations to several key related works.**

**Questions:**

1. Equation 2 uses ground-truth labels from the test set for model optimization. Does this violate standard protocol? Typically, ground-truth labels are used only for evaluation, not for model training or adaptation.
2. Can X-VLM be combined with other test-time adaptation methods to further improve performance? In Table 1(a), only results for X-VLM + Ours are shown, and comparisons with other related methods are missing.

**Details Of Ethics Concerns:**

No Ethics Concerns.

---

> ### Author Response · Authors · 2025-11-24
>
> **Thank you very much for your comprehensive review. We sincerely appreciate your keen attention to detail, especially regarding the logical flow, terminology, and contextual application of our work. We have made revisions and added new experimental results to address these concerns.**
>
> ## W1: Logical Flaw and unaccessible to online queries real-world scenarios.
> **A1**: We sincerely thank the reviewer for rigorously questioning the practical assumptions of our methodology. We acknowledge that our current evaluation protocol, which adapts over the entire test set (target domain), does not align with the fully online, per-query adaptation originally established by methods like TENT.
>
> However, we argue that the "Pretrain-then-Adapt" paradigm we propose addresses a distinct and highly valuable deployment scenario:
>
> 1. Addressing Unlabeled Deployment: Our paradigm is specifically designed for rapid, cost-effective deployment when a strong pre-trained retrieval model is available, but expensive human labeling for the new target domain is prohibitive.
>
> 2. The Deployment Scenario: We envision a scenario where, prior to actual online use, a manageable batch of unlabeled target data is collected once to perform the domain adaptation. The model is then deployed in this adapted state.
>
> 3. Benefits over Fully TTA: This approach offers significant advantages over fully online, per-query TTA:
>
>     -  Avoids Catastrophic Forgetting: Our method bypasses the constant risk of instability and catastrophic forgetting associated with sequential per-query adaptation (fully TTA requires frequent weight resets after multiple adapted to one query or batch).
>
>     - Cost Efficiency: It eliminates the high cost of data annotation.
>
> Therefore, "Pretrain-then-Adapt" is not an attempt to compete with the operational constraints of fully TTA, but rather a novel, application-focused paradigm providing a pragmatic solution for unlabeled domain transfer that is distinct from both traditional UDA (too costly) and online TTA (too unstable).
>
> ## W2 & W5: Terminology inconsistency, lack of citations.
> **A2**: We apologize for the lack of consistency and clarity in the initial submission. We have standardized the terminology throughout the paper to use "text-based person search" for consistent to the title. Then we have reviewed the experimental section and added necessary citations to relevant key related works to ensure proper context.
>
> ## W3: Lack of unique discussion on the text-based person retrieval task; method appears applicable to general vision-language tasks.
> **A3**: We agree that our method's core mechanism has potential applicability to general cross-modal tasks. However, we clarify our focused scope based on practical necessity:
>
> - Focused Application: We specifically focus on the Person Re-ID domain due to its unique practical constraints (e.g., privacy protection, making central data aggregation for UDA challenging) and deployment needs (e.g., federated learning or edge deployment), which strongly necessitate a resource-minimal TTA solution.
>
> - Contrast with General Retrieval: General vision-language retrieval often has more flexibility for extensive offline fine-tuning or UDA on public datasets. Our method is positioned as a critical solution for domains where such data aggregation or heavy training is prohibited. While our mechanism may be applicable elsewhere, its unique justification lies in addressing the constraints of person Re-ID.
>
> ## Q1: Does Equation 2 use ground-truth labels from the test set?
> **A4**:  Thank you for highlighting this ambiguity. We understand the concern that using ground-truth labels for optimization violates standard protocol. We clarify that **Equation 2** does not use ground-truth labels, where $y$ is the ground-truth label denoting wether $i$ and $t$ is truely-matched or not, however $p_{t2i}(y|t,i,\theta)$, $p_{i2t}(y|t,i,\theta)$ denote the probabilities of text-to-image and image-to-text search predictions. We also revised the surrounding text (marked in blue) in **Subsection Bidirectional Retrieval Disagreement as Uncertainty Proxy** to explicitly clarify that the loss construction relies only on model predictions, not external labels.
>
> ## Q2 & W4: Missing comparison with other TTA methods; lack of discussion on applying TTA to this task.
> **A5**: We recognize the necessity of positioning our work clearly against existing TTA methods. After directly adapting other TTA
>  methods to the text-based retrieval task, we update **Table 1 and subsection Comparison with other TTA methods** in the main paper to include direct comparisons of X-VLM combined with other established TTA methods marked in blue. These results explicitly demonstrate that standard TTA approaches can be applied to cross-modal retrieval, but often yield minor or negative gains, while our UATTA addresses the failure modes of generic entropy minimization losses (over confident to false negatives) in the cross-modal retrieval context.

---

### Official Review · Reviewer_vBDd · 2025-10-31

**Soundness:** 3
**Presentation:** 3
**Contribution:** 2
**Rating:** 4
**Confidence:** 3

**Summary:**

This paper proposes Pretrain-then-Adapt, an innovative framework designed for uncertainty-aware test-time adaptation (UATTA) in text-based person retrieval. The authors aim to address the domain shift problem, which is often observed in models trained on synthetic data but deployed in real-world environments. The method uses unlabeled test data at test time to adapt the model with minimal computational overhead. The core idea is to estimate prediction uncertainty through a bidirectional retrieval disagreement mechanism, which allows the model to recalibrate itself without relying on labeled data. The paper evaluates this approach across multiple benchmarks, showing improvements over traditional Pretrain-then-Finetune methods and existing test-time adaptation strategies.

**Strengths:**

1. The idea of uncertainty-aware adaptation for test-time adaptation (TTA) in text-based retrieval is interesting and addresses a significant gap in existing research. The use of unlabeled test data to adapt models in an efficient manner is a valuable contribution to practical deployment scenario.

2. The framework is designed for real-world applications where fine-tuning on labeled data is infeasible. This makes it a promising alternative to traditional pretrain-then-finetune paradigms.

**Weaknesses:**

1. The confidence filtering mechanism used in the framework is rather simple and somewhat “tricky”. It relies on basic engineering methods, such as disagreement between text-to-image and image-to-text retrieval. While effective, it does not appear to bring significant methodological innovation to the table. This aspect could be seen as a relatively basic engineering approach rather than a novel contribution to the field.

2. Despite its efficiency, the proposed method shows comparatively larger performance gaps when evaluated against fine-tuning-based methods, making it difficult to validate its effectiveness.

**Questions:**

See weakness

---

> ### Author Response · Authors · 2025-11-24
>
> **Thank you very much for your thoughtful feedback, particularly your comments on the novelty of the mechanism and gap against finetuning methods. We have clarified the principled distinctions and repositioned our task setting to address these concerns.**
>
> ## W1: The confidence filtering mechanism is simple and lacks significant methodological innovation.
> **A1**: We acknowledge the reviewer's perspective regarding the apparent simplicity of using bidirectional retrieval disagreement. However, we respectfully clarify the methodological novelty and principled nature of this design within the context of Test-Time Adaptation (TTA):
>
> - Principled TTA Mechanism: Our bidirectional retrieval disagreement proxy is not a generic filtering mechanism but a novel, task-specific signal designed to address the unique failure mode of TTA in cross-modal retrieval—namely, the tendency of entropy minimization to generate overly confident, yet incorrect, cross-modal matches.
>
> - Contrast with Prior Art: While cross-modal consistency is known, using the disagreement as an on-the-fly TTA uncertainty signal for feature adaptation is a novel concept in TTA literature. The strong performance gain of $\mathcal{L}_{\text{UATTA}}$ over existing retrieval TTA methods (like TCR), as shown in our ablation study (**Table 1 & 2 and Subsection Comparison with other TTA methods** in the main paper), validates that this principled design is highly effective and superior to basic engineering approaches.
>
> ## W2: Larger performance gaps against fine-tuning-based methods, difficult to validate effectiveness.
> **A2**: We appreciate the reviewer's focus on performance metrics and agree that fine-tuning (FT) results provide a crucial reference point. However, we clarify that comparing our TTA method directly against full fine-tuning without considering the deployment paradigm is misleading, as reasons below:
>
> 1. TTA vs. Fine-Tuning Paradigms: Fine-tuning methods (which serve as the upper bound for performance) require access to labeled target training data (or extensive pseudo-label generation) and involve massive, costly offline training to adjust all model parameters.
>
> 2. Our Context: UATTA operates under the stringent constraint of zero target labels and zero offline training, adapting to the target domain with limited access to pretrain model and test set only. This framework is designed for entirely different, highly practical scenarios, such as: federal deployment, expensive annotations and rapid transfer of applications
>
> Given these constraints, the objective is not to surpass the performance ceiling of FT, but to demonstrate the maximal improvement achievable under zero-resource adaptation. Our results show that UATTA significantly closes the gap between the unsupervised baseline and the supervised upper bound while maintaining unparalleled efficiency and flexibility.

---

### Official Review · Reviewer_P9ac · 2025-11-04

**Soundness:** 2
**Presentation:** 3
**Contribution:** 2
**Rating:** 2
**Confidence:** 4

**Summary:**

This paper proposes a new paradigm for text-based person retrieval called “Pretrain-then-Adapt,” which replaces the traditional “Pretrain-then-Finetune” approach. The core contribution is an Uncertainty-Aware Test-Time Adaptation (UATTA) framework that allows models to adapt to new target domains using only unlabeled test data, without requiring labeled target-domain data. It offers a label-free, uncertainty-driven test-time adaptation method that enhances the robustness and deployability of text-based person retrieval systems in real-world scenarios.

**Strengths:**

1. This paper introduces a label-free "Pretrain-then-Adapt" framework, eliminating the need for annotated target data.

2. This paper proposes a simple yet effective bidirectional retrieval disagreement to reliably detect uncertainty and prevent overconfidence.

3. The experiments demonstrate some improvements across multiple models and benchmarks with minimal computational cost.

**Weaknesses:**

1. The proposed "Pretrain-then-Adapt" paradigm appears functionally similar to unsupervised domain adaptation (UDA) or unsupervised learning for text-based ReID. The core distinction—adapting only a few parameters (LN layers) at test time—is an incremental engineering contribution rather than a foundational shift in paradigm, as claimed.

2. The core innovation, using bidirectional retrieval disagreement as an uncertainty signal, is a straightforward application of cross-modal consistency. This concept is a well-established principle in cross-modal and unsupervised learning. The method essentially uses this consensus as a pseudo-label quality signal, an approach that is not novel in the broader machine learning context.

3. The empirical evaluation lacks critical comparisons with state-of-the-art unsupervised text-based ReID methods. This omission makes it difficult to assess the true contribution of UATTA versus simply applying established unsupervised learning techniques to the test set.

4. The paper relies solely on quantitative results. Qualitative visualizations (e.g., t-SNE plots of features before/after adaptation, examples of successful/failed adaptations) are missing. Such analysis would greatly strengthen the claim of mitigating domain shift and provide intuitive insights into the method's behavior.

5. The theoretical derivation linking parameter variance to bidirectional disagreement, while a nice addition, relies on strong assumptions (e.g., symmetric consistency for an ideal model) and first-order approximations. A more rigorous analysis or an ablation on the impact of this specific theoretical motivation is needed.

**Questions:**

What is the fundamental conceptual and methodological distinction between the proposed "Pretrain-then-Adapt" paradigm and standard Unsupervised Domain Adaptation (UDA) or unsupervised learning for text-based ReID? The process of adapting a pre-trained model using unlabeled target data appears to align closely with the core objective of UDA. Please clarify the novel theoretical or practical contribution of the paradigm itself, beyond the specific mechanism of UATTA.

For other questions, please refer to the points raised in the Weaknesses section.

---

> ### Author Response · Authors · 2025-11-24
>
> **Thank you very much for your time and comprehensive review. We sincerely appreciate your rigorous assessment, which highlights crucial points regarding our paradigm's novelty, mechanism, and empirical scope. We have conducted significant revisions and additional experiments, including a new comparison table and extensive qualitative analysis, to address all concerns.**
>
> ## Q & W1 & W3: The "Pretrain-then-Adapt" paradigm is functionally similar to UDA, and the TTA focus is an incremental engineering contribution. Lackness of comparison to UDA methods.
>
> **A1 & A3**: We appreciate the reviewer's query regarding the fundamental novelty of the "Pretrain-then-Adapt" paradigm, especially its relationship to Unsupervised Domain Adaptation (UDA). We clarify the crucial distinctions:
>
> 1. Fundamental Distinction from UDA
>
> The proposed "Pretrain-then-Adapt" paradigm, specifically implemented via Test-Time Adaptation (TTA), is fundamentally distinct from standard Unsupervised Domain Adaptation (UDA) in its scope, resource requirement, and deployment stage:
>
> - UDA (Traditional): UDA methods typically require access to the entire unlabeled target training set during a distinct domain adaptation phase. This phase can take hours or days and is often performed offline.
>
> - Our Paradigm (TTA): TTA operates strictly on test samples immediately before inference. It requires no access to a target training set, no offline training, and only pretrained model weight and test data.
>
> This distinction is not merely an "incremental engineering contribution"; it is a practical paradigm shift for deployment scenarios (like edge devices or privacy-sensitive applications) where performing heavy offline training is infeasible. The core novelty lies in designing a stable, effective self-supervision mechanism ($\mathcal{L}_{\text{TTA}}$) that works robustly under the highly limited federated deployment (limited access to pretrained model weight) and compute budget (few time limit) of the TTA constraint.
>
> 2. Novelty in Unsupervised Context
>
> To clarify the contribution within the broader unsupervised context, we have updated **Table 2** and a **Subsection: Comparison with other Semi/Un-supervised Methods** in the main paper to include comparisons with state-of-the-art Unsupervised (UDA) and Weakly-Supervised Text-Based ReID methods. The results demonstrate that our TTA approach achieves comparable or superior performance to these complex UDA methods without needing their extensive target training data or multi-stage pseudo-labeling pipelines.
>
> ## W2: The core innovation, using bidirectional retrieval disagreement as an uncertainty signal, is a straightforward application of cross-modal consistency and is not novel.
>
> **A2**: We acknowledge that cross-modal consistency is a well-established principle in multimodal learning. However, we respectfully think our method is different to previous TTA method. Prior Test-Time Adaptation research primarily relies on entropy minimization or diversity regularization within a single modality or regarding modality interaction as a entirety. Our work is the first to explicitly leverage the bidirectional asymmetry of the cross-modal retrieval task (text-to-image vs. image-to-text) to derive an uncertainty signal ($\mathcal{L}_{\text{UATTA}}$) that guides TTA. This signal is designed to address the specific failure mode of TTA in retrieval: over-confidence on false matches induced by simple entropy minimization.
>
> ## W4: Qualitative visualizations (e.g., t-SNE plots, successful/failed adaptations) are missing.
> **A4**: We fully agree that qualitative visualizations are crucial for intuitive understanding and demonstrating the mechanism's effectiveness. We apologize for the omission in the initial submission due to space constraints.
>
> We have now added extensive qualitative analysis in the Supplementary Material:
>
> - t-SNE Visualization (**Section B.1 and Figure B.1**): We include a t-SNE plot that explicitly visualizes the feature space before and after TTA. This visually confirms that UATTA successfully reduces the domain gap by aligning Query features with Gallery clusters.
>
> - Retrieval Examples (**Section B.2 and Figure B.2**): We provide detailed qualitative comparison examples showing successful adaptations (mitigating over-confidence) and examples where UATTA successfully resolves fine-grained semantic confusion that the baseline failed to address.
>
> These visualizations provide strong, intuitive support for our claims regarding domain shift mitigation and the robustness of the score distribution.

---

> > ### Author Response · Authors · 2025-11-24
> >
> > ## W5: The theoretical derivation relies on strong assumptions and first-order approximations. More rigorous analysis is needed.
> > **A5**: We appreciate the reviewer's scrutiny of the theoretical underpinning, particularly concerning the necessary assumptions for the derivation.
> >
> > We acknowledge that the derivation relies on certain working assumptions (e.g., local symmetry in feature space consistency) and uses first-order approximations typical for practical optimization objectives. Our intent was to provide a principled motivation for linking the empirical bidirectional disagreement to the theoretical goal of minimizing parameter variance ($\sigma^2$), rather than providing a formal mathematical proof of convergence.
> >
> > - We propose to revise the phrasing in the main paper to reflect this as a "motivated objective" or "intuitive explanation" rather than a formal, strict "derivation." This adjustment aims to manage expectations and clearly state that the purpose is guiding the design of the loss function, $\mathcal{L}_{\text{UATTA}}$.
> >
> > - We will strengthen the discussion in the main paper's **Subsection: Comparison with other TTA methods and Table 1**  as marked in blue to clearly illustrate how the differing assumptions about uncertainty (our cross-modal disagreement vs. single-modal entropy e.g., Tent) lead to distinct practical benefits in feature adaptation.
> >
> > We are sincerely open to any specific suggestions you may have for refining the theoretical formulation or structure to better address your concerns.

---

### Official Review · Reviewer_AFxn · 2025-11-04

**Soundness:** 2
**Presentation:** 2
**Contribution:** 2
**Rating:** 4
**Confidence:** 3

**Summary:**

This paper proposes UATTA (Uncertainty-Aware Test-Time Adaptation), a novel framework for text-based person retrieval that eliminates the need for labeled target-domain data. The authors introduce a new paradigm — Pretrain-then-Adapt, replacing the traditional Pretrain-then-Finetune pipeline. The key idea is to leverage Bidirectional Retrieval Disagreement (BRD) as an uncertainty measure between text-to-image and image-to-text retrieval rankings. This uncertainty guides test-time entropy minimization to avoid overconfident false positives during adaptation.The method is theoretically grounded through an analysis showing proportionality between parameter variance and retrieval disagreement, and empirically validated on four datasets (CUHK-PEDES, ICFG-PEDES, RSTPReid, and PAB) using both CLIP and XVLM frameworks. UATTA improves retrieval accuracy while requiring minimal computational cost and no labeled data.

**Strengths:**

1. The proposed Pretrain-then-Adapt pipeline directly addresses the impracticality of relying on labeled target-domain data in text-based person retrieval. This design is timely and pragmatic for real-world deployment under privacy and resource constraints.

2. Extensive evaluations on several benchmarks confirm consistent gains, and the proposed method can achieve competitive results while requiring only 0.08 GPU-hours for adaptation. The ablation studies are thorough and insightful.

**Weaknesses:**

1.This paper adopts many hand-crafted parameters and designs, like K and uncertainty formulation. Would these designs affect the generalization performance across different models?

2.The authors only update LayerNorm affine parameters to ensures stability, but it might limit adaptability. Hence, could lightweight tuning modules (e.g., LoRA, adapters) provide a better performance–efficiency tradeoff?

**Questions:**

see weakness

---

> ### Author Response · Authors · 2025-11-24
>
> **Thank you very much for taking the time to review and for your support. We try our best to address your questions as follows.**
>
> ## W1: The generalization performance of parameters and designs across different models.
> **A1**: Thank you for raising this important question regarding the stability and generality of our design choices. We appreciate this constructive question regarding the stability and generalizability of our proposed designs. We agree that rigorous justification for design parameters is essential.
>
> 1. Rationale for Parameter $K$
>
> We clarify the principled selection of $K$, which defines the mutual top range for bidirectional retrieval. While we found the empirically optimal result at $K=3$, we intentionally set the default $K=5$ because the RSTPReid dataset contains $\mathbf{5}$ ground-truth images per query identity. This choice is deliberate, as $K=5$ logically represents the boundary between true and false positives in the ranking list, providing a principled and non-ad-hoc justification for our setting. We have detailed the impact of different $K$ values in **Section A.2** and **Table A.3 of the Supplementary Material**.
>
> 2. Generalization Across Architectures
>
> Regarding generalization and hand-crafted designs, our core designs, including the uncertainty formulation and the bidirectional retrieval proxy, are agnostic to the backbone architecture, as mentioned in **Section 3 Method Problem Formulation**, **Section 4 Experiment Setting** and **Table 3 in the main paper**. Experiments on two representative architectures suggest that our approach introduces robust training objectives rather than model-specific ad-hoc adjustments, which inherently supports the generalization and broad applicability of UATTA.
>
>
> ## W2: More experiment of ightweight tuning modules.
> **A2**: We thank the reviewer for suggesting this insightful comparison with Parameter Efficient Fine-Tuning (PEFT) methods, which are critical for deploying large models efficiently. We have conducted additional experiments comparing UATTA with representative PEFT methods, including Prompt Learning, Prefix-Tuning and LoRA, integrated with our test-time adaptation loss. The detailed results are presented in **Section A.1** and **Table A.1 of the Supplementary Material**.
>
> The results consistently show that PEFT methods fail to maintain or improve performance under our test-time adaptation setting. This outcome can be attributed to the nature of the adaptation objective:
>
> 1. PEFT methods primarily focus on efficiently adjusting parameters for supervised fine-tuning (SFT) tasks where explicit labels guide the learning process.
>
> 2. The entropy minimization objective, which is label-free, does not provide sufficient, high-quality supervisory signals necessary to learn discriminative representations by solely adjusting the small set of trainable PEFT parameters.
>
> This confirms the necessity of our uncertainty-aware full-model adaptation strategy to effectively align cross-modal features during TTA.

---

### Meta-Review · Area_Chair_16a5 · 2026-01-05

**Summary:**

The paper introduces "Pretrain-then-Adapt," a paradigm for text-based person retrieval designed to replace traditional fine-tuning with Uncertainty-Aware Test-Time Adaptation (UATTA). This framework utilizes a bidirectional retrieval disagreement mechanism to estimate uncertainty and guide model adaptation using only unlabeled test data.

**Reviewer Concerns:**

1. Reviewer P9ac argued that "Pretrain-then-Adapt" is functionally similar to UDA or transductive learning. While the authors argued the distinction lies in the resource constraint (no target training set, limited compute), the reviewer viewed this as an incremental engineering contribution rather than a fundamental paradigm shift.
2. Reviewer aw8W noted that real-world retrieval is typically sequential/online, whereas this method requires a batch of test data. The authors conceded their method is not fully online (per-query) but argued it fits a "federated/edge deployment" scenario. This confirms the method does not meet the strict definition of streaming Test-Time Adaptation, limiting its claimed versatility.
3. Reviewers vBDd and P9ac felt the bidirectional retrieval disagreement was a "straightforward application of cross-modal consistency" rather than a deep methodological innovation.

**Reviewer Scores:**

The reviewers acknowledged the practical motivation of eliminating the need for labeled target-domain data, which is valuable for privacy-sensitive applications. However, significant concerns were raised regarding the fundamental novelty of the paradigm relative to Unsupervised Domain Adaptation (UDA), the applicability of the batch-adaptation setting to real-world "online" retrieval scenarios, and the lack of qualitative analysis in the initial submission. The authors provided a rebuttal that included new comparisons with Parameter-Efficient Fine-Tuning (PEFT) methods (e.g., LoRA) and other TTA approaches, as well as qualitative visualizations (t-SNE). While they clarified that their method operates under stricter resource constraints than traditional UDA (no access to a target training set), the reviewers' core concerns about the incremental nature of the contribution and the "batch" vs. "streaming" distinction remain points of contention.

---

### Decision · Program_Chairs · 2026-01-26

Reject